# Planar Cell Polarity Signaling: Coordinated Crosstalk for Cell Orientation

**DOI:** 10.3390/jdb12020012

**Published:** 2024-04-29

**Authors:** Sandeep Kacker, Varuneshwar Parsad, Naveen Singh, Daria Hordiichuk, Stacy Alvarez, Mahnoor Gohar, Anshu Kacker, Sunil Kumar Rai

**Affiliations:** 1Department of Pharmacology, Medical University of the Americas, Charlestown KN 1102, Saint Kitts and Nevis; s.kacker@mua.edu; 2Department of Human Body Structure and Function, Medical University of the Americas, Charlestown KN 1102, Saint Kitts and Nevis; v.parsad@mua.edu (V.P.); d.hordiichuk@mua.edu (D.H.); 3Department of Cerll and Molecular Biology, Medical University of the Americas, Charlestown KN 1102, Saint Kitts and Nevis; n.singh@mua.edu (N.S.); s.alvarez@mua.edu (S.A.); m.gohar@mua.edu (M.G.); 4Department of Histology and Human Physiology, Medical University of the Americas, Charlestown KN 1102, Saint Kitts and Nevis; a.kacker@mua.edu

**Keywords:** planar cell polarity (PCP), tissue morphogenesis, neural tube defects (NTDs), Wnt signaling, Notch signaling, Hippo signaling, negative regulation

## Abstract

The planar cell polarity (PCP) system is essential for positioning cells in 3D networks to establish the proper morphogenesis, structure, and function of organs during embryonic development. The PCP system uses inter- and intracellular feedback interactions between components of the core PCP, characterized by coordinated planar polarization and asymmetric distribution of cell populations inside the cells. PCP signaling connects the anterior–posterior to left–right embryonic plane polarity through the polarization of cilia in the Kupffer’s vesicle/node in vertebrates. Experimental investigations on various genetic ablation-based models demonstrated the functions of PCP in planar polarization and associated genetic disorders. This review paper aims to provide a comprehensive overview of PCP signaling history, core components of the PCP signaling pathway, molecular mechanisms underlying PCP signaling, interactions with other signaling pathways, and the role of PCP in organ and embryonic development. Moreover, we will delve into the negative feedback regulation of PCP to maintain polarity, human genetic disorders associated with PCP defects, as well as challenges associated with PCP.

## 1. Introduction

The process of cell polarization during embryonic development is crucial for identifying the specific domain in both vertebrates and invertebrates [1]. Cell or tissue polarization occurs more frequently in epithelia that display two kinds of polarity: planar cell polarity (PCP) and apical–basal polarity [2]. The concept of PCP refers to coordinated cell orientation within cells and epithelia that affects cellular activities that result in changes in tissue shape, reflecting the importance of PCP for the proper functioning and development of various organs or tissues. PCP is not only restricted to epithelial cells but also occurs throughout an animal’s whole developmental window in mesenchymal cells (MSCs) [3]. Core PCP and Ft-Ds-Fj (Fat–Dachsous–Four-jointed) signaling pathways were explored in Drosophila as regulators for the coordinated cell orientation of external hairs and bristles [4,5,6,7,8]. PCP effectors, including Fuz, Intu, and Wdcp, are essential elements of the pathways involving planar cell polarity [9]. Coordinating cell polarization within an epithelium’s plane is critical for several developmental processes as well as tissue structure, and this route makes this possible. The essential elements of the PCP pathway, which is shared by all species, are the proteins Vang-like (vang), Frizzled (Fz), Dishevelled (Dvl), and Prickle (Pk), and these proteins aid in the development of trichomes and the determination of the proximal and distal cell fates in wing cells in Drosophila [10,11]. The PCP pathway has been linked to the formation of epidermis, hair follicles, and inner ear stereociliary bundles in vertebrates [12]. It can have an impact on left–right patterning and cilia function, and it is also involved in renal development [13]. Planar polarization is characterized by the mutually exclusive and complementary distribution of molecular signaling complexes in the trans-membrane that leads to uneven abundance in different cell organelles in every cell in mosaic tissue; this asymmetrically patterned tissue promotes the coordinated orientation of sub-cellular structures and cellular behaviors through regulating cellular adhesion and cytoskeletal components. Maintenance and establishment of proper plane polarity across the tissue during the developmental window is a crucial process that is involved in the repair and homeostasis of tissue, and abnormalities in PCP signaling have been linked to a variety of clinical diseases. The goal of this review paper is to present a thorough overview of the molecular mechanisms behind PCP signaling, its history, its constituent parts, its connections with other signaling pathways, and its involvement in the development of organs and embryos. We will also discuss challenges associated with PCP, human hereditary disorders associated with abnormalities in PCP, and the negative feedback control of PCP to maintain polarity.

## 2. Historical Notes

In the past, research on PCP began with arthropods, and Drosophila melanogaster was used to study PCP’s function [14]. Among the most often-examined locations for PCP in Drosophila are the eye, abdomen, and notum of the insect [15,16,17,18]. PCP is associated with the lgl (lethal giant larval) gene, of which Bridges and Brehme initially reported a mutant allele in 1944 [19,20]. Up until 2000, it was still believed that there were just three mutations in this class of Drosophila [21,22]. Planar polarity and its transmission between separate cells were first described in 1975 by Lawrence and Shelton’s groundbreaking work [22,23,24]. Studies conducted on fruit flies have revealed that two major signaling pathways—a set of "core" PCP components, including Diego, Vang (Van Gogh), Fmi (Flamingo), Pk (Prickle), Fz (Frizzled), Dsh (Dishevelled), and Ft-Ds-Fj (Fat–Dachsous–Four-jointed) modules—are responsible for PCP [25,26]. The epidermis/skin and inner ear of mammals, as well as body bristles and wing hairs in insects, are prime examples of epithelial cells possessing PCP characteristics [27].

## 3. Molecular/Cellular Mechanisms—PCP Signaling Pathway

A growing body of research links dysregulated Wnt/PCP signaling to cancer as a basic developmental route. The fact that developmental Wnt/PCP signaling is crucial for both tissue polarity and cell migration means that the hijacking of this system to accelerate tumor progression is not surprising. Cell orientation is determined by local variations in Fz cellular activity rather than threshold concentrations of Fz within each cell [28]. As a result, a well-established model was created, according to which the gradient cellular activity of Fz for graded ligand responses promotes the appropriate direction for planar polarity. According to another model [28], cells are dependent on local head-to-tail alignment through the unequal distribution of PCP signaling pathway components. These two hypotheses provide opposing predictions about the roles of global spatial signals that determine the location of Fz localization and provide differential patterning of Fz activity, respectively, either at the tissue or cell level. Coordinating or antagonistic interactions occur between all the components of the PCP signaling pathway. Mechanisms underlying PCP signaling were explored in the Drosophila model. Morphogen gradients direct embryonic axes, coordinate morphogenetic motions, and instruct cell divisions and organ growth at the tissue or organism level. At the cell level, communication between cells is stimulated by core PCP factors that facilitate polarity coordination between cells, either indirectly or directly via both extracellular and embryonic cues. There are two evolutionary conserved signaling modules, i.e., the core PCP and Ft-Ds-Fj modules, that coordinate with each other to establish the proper PCP [9]. Ds (Dachsous) and Ft (Fat) encode global polarity signals that are further perceived by the asymmetric buildup of cell–surface complexes followed by cue transmission across surrounding cells, in conjunction with Fj (Four-jointed), a trans-membrane Golgi complex protein [14,29,30,31]. In Drosophila wing cells, Vang, Pk, Fz, Dsh, and Dgo accumulate at the proximal and distal sides of the apical-cell membrane, respectively, whereas Fmi, which is distributed at both the distal and proximal edges of the cell, acts as a molecular bridge due to the formation of Fmi homodimer across the apical cell to cell junctions [2,32,33,34,35]. Core PCP factors move distally via apical proximodistal microtubule arrays after endocytosing Fmi-Fz complexes, resulting in an asymmetric localization inside cells [36,37]. Moreover, cell-to-cell contact is necessary for the formation of these intracellular asymmetries and the distribution of planar polarity in the plane of the epithelium. In particular, dominating non-autonomy events arise when cell clones exhibit genetic ablation in key PCP molecules such as Fz and Vang, leading to directionally altered and damaged PCP in the surrounding normal cells [38]. Additionally, Vang or Fz gain of function mutant cells reorient the distantly located wild-type surrounding cells away from clones, whereas Fz loss of function mutant cells orient the neighboring wild-type cells toward clones [17]. These findings suggest that planar polarity may be spread within the epithelium along a gradient of Fz cellular activity and that regional glitches in PCP signaling cause an alteration in regional symmetry. Extracellular interactions between Vang-Fmi and Fmi-Fz complexes facilitate the transmission of PCP cues between adjacent cells [35,37]. Studies using genetic, molecular, and computational analysis revealed that an embryonic symmetry disruption event or global directional signal is translated into graded Fz cellular activity. This is then enhanced by the activity of PCP core molecules at cell junctions via a feedback-loop mechanism [35,39,40]. In Figure 1, PCP-mediated signaling is depicted. The specific chemical mechanisms underlying the PCP signaling pathway remain unknown despite numerous investigations.

## 4. Core Components

Studying the functions of the many components in the PCP signaling pathway is crucial (Table 1). Van Gogh (VANGL in vertebrates), Fz (FZD in vertebrates), Dsh (DVL in vertebrates), and Dgo (ANKRD6 in vertebrates) are the six core proteins of Fz/PCP (Fz-Vangl/PCP signaling in vertebrates) signaling processes that interact with other intra- and intercellular proteins (Table 1). Extensive insights into the regulation of PCP have been gained from analyses conducted on Drosophila and many model organisms that are vertebrates. Through genetic investigations of Drosophila, a major conserved pathway regulating PCP was first discovered. This pathway is known as the core Wnt–Frizzled PCP (Fz/PCP) signaling pathway. Research on PCP in vertebrates—most notably mice and zebrafish—has revealed cellular characteristics that need input from PCP signaling and found novel PCP signaling components. The role of Van Gogh (Vang)/Vangl genes in this molecular system has become the main focus of these studies.

## 5. PCP Complex

Two different PCP complexes on opposing domains of each cell communicate to form the planar orientation axis. Dsh, Dgo, and Pk comprise the cytoplasmic components, while the remaining three factors (Fmi, Fz, and Van Gogh) are trans-membrane components. PCP signaling trans-membrane components help transfer polarity information between cells (Figure 2), while its cytoplasmic components aid in intracellular asymmetry and the conversion of polarity cues into cellular activity [4,5,6,7,8]. A homophilic adhesion mechanism is thought to be the role of Fmi [42]. It is also referred to as a molecular PCP interaction partner because it co-localizes with both Stbm/Vang-Pk and Fz-Dsh complexes. In addition to bringing Pk to the plasma membrane by Stbm (Vang), Pk interacts with Dsh and inhibits its plasma membrane recruitment by Fz [43,44]. Vang brings Dsh and Pk to the plasma membrane after interacting with these factors and prevents the antagonistic effects of the Pk-Stbm complex by directly interacting with Dsh [45,46,47]. According to evidence from genetic ablation experiments, the pairs Stbm/Vang-Pk and Fz-Dsh are thought to be hostile, leading to their localization at opposite poles of each cell [42].

## 6. Role of Cell Adhesion Molecules

Numerous aspects of cell polarity are regulated by cell–cell adhesion receptors, like front–rear polarity in migrating cells, PCP during the development of organs, and apicobasal polarity in endothelial and epithelial cells [48]. The first indication that PCP and the cell adhesion molecule (CAM) are closely related came from the discovery of Fmi (atypical cadherin) [33]. An unusual transmembrane cadherin called Fmi is found in both the proximal and distal regions of the cell before the development of PCP symptoms, and it is controlled by the protein Fz. Ft and Dsh are the other two CAMs that play a crucial role in PCP [48,49,50,51,52]. Genetic experiments conducted on Drosophila revealed that Dsh and Fat are involved in the PCP signaling pathway upstream of Fmi and other Fz complex molecules that are involved in the PCP signaling pathway [49,50,51]. Ft is a 560 kDa protein with 2 laminin G domains, 5 EGF-like domains, 34 cadherin repeats, and a cytoplasmic domain unique to cadherins encoded by Ft genes, whereas Dsh, encoded by Dsh genes, has a unique cytoplasmic domain and 27 cadherin repeats. In an experimental study of genetic ablation, it was found that Dsh negatively affected the activity of Fat and that it functions upstream of Fat [50]. Further studies based on biochemical tests, which demonstrated that it interacted with Dsh but not with homophilic adhesion pathways, supported this [52]. In the wing and eye, Dsh is expressed as a gradient, which conveys polarity information [50,53], while PCP is disrupted by Dsh loss but can be restored by ubiquitous expression of Ds [53,54]. In the PCP signaling pathway, Ft negatively regulates downstream genes through its binding to atrophin, a co-repressor found in the nucleus [55], which is evident from the atrophin mouse model.

## 7. Tissue Morphogenesis—Planar Cell Polarity

Most people agree that the dynamics of epithelial tissue, which lines most of our organs, drive morphogenesis. PCP systems can be used to arrange polarity in motile populations undergoing dynamic morphogenetic changes and non-motile tissues where cells maintain contact with a stable set of adjacent neighbors. A conserved developmental process called body axis elongation that results from systematic processes of cell division, shape change of cells, cell motility, rearrangement of cells, and cell divisions, requires spatial information and directional signals from cell fate specification along the AP axis and PCP signaling pathways, respectively [28]. There has been a decade of research on how the equivalent Fz/PCP signaling system is responsible for the regulation of convergent-extension movements in Danio rerio, zebrafish, Xenopus laevis, and frogs [55,56,57,58]. The involvement of Fz/PCP signaling in vertebrate tissue development/morphogenesis has been demonstrated by ongoing studies. Gastrulation involves several well-defined polarized cell orientations, including convergent extension motions that constrict and elongate the developing germ layers antero-posteriorly [59]. An example of such behavior occurs when ML (mediolateral) elongated MSCs intercalate between their anterior and posterior neighbors in a similar cell layer through polarized planar intercalation. Consequently, tissues are elongated along the AP axis and narrowed along the ML axis [60]. Additionally, polarized radial intercalations and directed cell migration play a critical role in convergent extension, where cells prefer to intercalate with their deep and superficial neighbors, respectively [61]. Alteration in the convergent extension process reflects a wing-hair error in Drosophila, considered a characteristic feature of the impaired PCP signaling pathway. Impaired neural tube broadening in fish and frogs leads to malformed neural tubes occurring due to an altered PCP-dependent convergent extension process [62,63,64]. In mice, mutations in the PCP core component led to craniorachischisis that extends from the hindbrain to the tail. In humans, genetic defects in VANGL1/2 genes are associated with failure in neural tube closure [65,66]. Additionally, the improper orientation of follicles and coat hair shafts in mice, posterior displacement of MTOC (microtubule organizing center), and impaired stereocilia polarity are other examples of tissue dysmorphogenesis, which is also observed in PCP mutants [67,68,69,70,71,72]. PCP signaling is also involved in axonal projection guidance and cell division orientation in mice and zebrafish gastrula, respectively [73,74].

## 8. Neural Tube Closure

The process of the neural tube closing is complex and involves a number of cellular processes, including interkinetic nuclear migration, apical constriction, and convergent extension. Furthermore, distinct molecular mechanism regulation through the transcription factors Grhl2/3, Pax3, Cdx2, and Zic2, the non-canonical Wnt/planar cell polarity pathway, and SHh/BMP signaling is needed [75]. Neural tube closure is driven by the motions of cells undergoing a convergent extension process and occurs along the rostral–caudal axis at three distinct locations. The prevalence of neural tube defect (NTD) among neonates in the US is estimated to be 0.001, or 0.002/1000 [76]. The NTD that affects the brain is not the same as the NTD that affects the spinal cord. NTDs can be classified into two groups: (i) Open defects, such as lipomyelomeningocele, exencephaly–anencephaly, and craniorachischisis. (ii) Encephalocele, split cord malformation, dermal sinus, spina bifida occulta, and dermal nevus are examples of closed abnormalities [77,78,79]. Partial closure of the spinal cord referred to as spina bifida is a more prevalent kind of human NTD. Studies based on family and population have shown that NTD is multi-factorial in origin [41,80,81,82,83,84]. A study performed on *Xenopus* demonstrated that the PCP signaling-mediated convergent extension process is needed to fill the gap between raising neural folds and permit their fusion and apposition [41]. Other studies based on mice, zebrafish, and *Xenopus* showed that PCP signaling plays a crucial role in the convergent extension process during neurulation as well as in the gastrulation process [54,56,59,81,82]. Previous research on patients with familial and sporadic NTDs revealed that NTD is possibly caused by genetic mutations in the Vangl1 gene, which is a component of PCP [41]. They also reported that the Vangl1 gene mutation obstructed the interaction between Dvl and VANGL1. The Ltap gene mutation results in caudal defects in *looptail* mice, like craniorachischisis [85]. In addition, some rostral abnormalities are also seen in Dvl2^−/−^ mice; caudal NTDs are thought to be caused by the PCP signaling cascade [86,87]. Defective cilia formation and open neural tubes occur in *Xenopus* embryos where PCP effector proteins Fuzzy and Inturned are depleted [88]. Dominant negative Dvl gene mutations lead to NTD by disrupting the apical constriction that is involved in the closure and bending of the neural tube [88]. Dominant negative Fz6^−/−^ and Fz3^−/−^ mutations lead to craniorachischisis with 100% penetrance, and mice with these mutations die shortly after delivery [86]. One study showed that genetic mutations in Fz2 or Fz1 result in NTD [87]. Scy and Crsh mutant mice (Celsr1 homozygous mutants) exhibit severe NTDs including craniorachischisis [89]. Trilobite mutant embryos lacking Stbm exhibit abnormalities in hindbrain neuron posterior migration and gastrulation movements, leading to NTDs [90]. Several studies have shown that ablating the Lpp1 gene in looptail mice results in serious NTDs, including craniorachischisis [85,91]. Ltap/Vangl2 and Wnt5a may interact genetically to increase NTD penetrance; as a result, mice carrying LtapLp/+ and Wnt5a^−/−^ mutations displayed craniorachischisis [92]. Vangl2 mutations were also observed in NTD fetuses that were miscarried or stillborn [70]. Animal model studies demonstrated that genetic ablation in Diversin (Diego) altered the convergent extension process in *Xenopus* and zebrafish [65,93]. Genetic studies showed that loss or gain of Pk1 function results in disruption in the convergent extension process leading to spina bifida [94,95].

## 9. Tissue Regeneration

Polarization of cells and tissues is an elemental process for tissue morphogenesis during regeneration and development. Studies have proved that PCP signaling pathways are involved in tissue regeneration [95,96,97,98]. PCP systems regulate various processes, including sensory organ formation, animal locomotion, directional tissue growth, tissue re-shaping, convergence extension, and orientation of actin filament and wing hair, during tissue regeneration and embryogenesis in multiple species [4,95,96]. A proper apical–basolateral polarity is essential for tissue regeneration in addition to being vital for the maintenance and performance of epithelial tissues [98]. During regeneration of the tadpole tail or during normal development, Vangl2 can also be lost, and excess neural tissue is produced, indicating that Vangl2 is involved in the regulation of nerve growth and regeneration [98]. According to one study, Vangl2 is expressed asymmetrically on sensory epithelial cells, and this expression pattern persists during regeneration and following ototoxic damage [99]. PCP mediates the regeneration of the spinal cord in axolotls by inducing the expansion of neural stem cells [96]. PCP is involved in the development of cone photoreceptors during the regeneration and growth of the retina in *Danio rerio* [100].

## 10. Developmental Process

PCP is involved in various developmental processes and acts as a steering wheel to control the activity of downstream polarized cell behaviors in response to external global stimuli. PCP can polarize a broad variety of cell behaviors, indicating that it may interact with multiple downstream effectors.
(i)**Convergent Extension Process:** It is the first process that is found to be associated with PCP [56]. MSCs stretch and produce mediolateral protrusions during the convergent extension process. These protrusions incorporate mediolaterally, restricting the mediolateral axis and lengthening the AP axis (Figure 3) [101]. Depletion of PCP components has been linked to mediolateral intercalation, polarization, and elongation, according to several experimental findings [54,56,59,102]. Only two discoveries provide direct mechanistic links between convergent extension behaviors and asymmetrically localized core components of PCP, even though many PCP-dependent mechanisms have been hypothesized to mediate convergent extension movements. On the A-P sites of intercalating cells in neuro-epithelial cells, PCP determines the region of myosin localization. Dvl and Fmi/Celsr1 recruit formin-DAAM1 to A-P sites, where it interacts with PDZ-RhoGEF, activates RhoA, and increases myosin contractility, bending the neural plate and mediating directed intercalation of cells [103]. A comparable mechanism has been observed to propel the convergent extension movements of MSCs during the gastrulation process of Xenopus laevis. In Xenopus laevis gastrulation, Dsh and Fritz induce localization of septin towards the mediolateral vertices, where they restrict junctional shrinkage and cortical contractility of cortical actomyosin spatially to the margins of A-P cell ends [104,105]. Collectively, these investigations demonstrate how spatial cytoskeleton modification resulting from asymmetric PCP localization leads to collectively polarized cell behaviors.(ii)**Positioning—Cilia and Centrosome:** PCP controls the orientation of microtubule-based structures such as cilia and the mitotic spindle by regulating the positioning of the mitotic spindle along the plane of epithelial cells through interaction with the SOC (spindle orientation complex), followed by binding of microtubules astral to the cell periphery with the help of dynein complex [106]. To orient the spindle posteriorly, astral microtubules and the dynein complex are brought to the posterior cortex through the interaction of Dsh with Mud/NuMA, and Mud/NuMA is recruited by Pins/LGN on the anterior side, which causes the spindle to orient A-P. A non-dividing inner ear cell’s kinocilium is orientated by PCP in conjunction with its spindle orientation machinery [107,108]. The mPins/LGN and Gai localize in vestibular hair cells to the abneural periphery, which is located across from Vangl2; they are necessary for the positioning of kinocilia, followed by subsequent stereocilia bundle alignment [107]. Dynein and the plus ends of microtubules also exhibit an abneural bias, indicating that Gai-mPins/LGN pull on microtubules through a process akin to that because it is responsible for centrosome positioning during spindle orientation. According to one study, Vangl2 is needed for Gai-Pins/LGN-crescent to properly align between cells that coordinate the positioning and polarity of kinocilia and stereocilia, respectively, throughout the tissue [107]. Studies have observed that PCP is needed for asymmetric positioning of cilia in a wide range of cells [67,68,69,109]. Hence, PCP determines both the plane of cell division in dividing cells and specifies cilia positioning in non-dividing cells.(iii)**Distal Positioning—Wing Hairs:** Every Drosophila wing blade cell has a distal end with an actin-rich protrusion. The locations of wing hair and the Fz-Dsh-Fmi positions are closely correlated, which implies that core proteins may be responsible for localizing cytoskeletal regulators to certain areas of cells [110]. A group of proteins known as Fuzzy, Fritz, and Inturned is recruited by Vang to the proximal junction, which negatively regulates the formation of actin pre-hairs [103,104]. Actin polymerization is thought to be repressed by Fuzzy, Fritz, and Inturned proteins by regulating multiple wing hairs, a GBD/FH (GTP-binding/formin homology)-3 domain protein [110,111,112]. As a result, actin nucleation occurs at distant positions within the cell, and ectopic actin bundles grow over the apical surface in the absence of multiple wing hairs [113]. The pre-hair nucleation process precedes distal nucleation, and casein kinase 1g CK1/gilamesh is required for further vesicle trafficking coordination with Rab11 [114]. Rho and Drok (Rho–kinase complex) also play a role in wing-hair formation, but the precise role of Rho is tedious to explore because of its involvement in various functions in cells, including cell division and cell shape [115,116].

## 11. Cochlea

The inner ear, notably the organ of Corti, is the clearest example in the vertebrate system where the link between PCP and cilia is well understood [117]. Assorted V-shaped stereociliary bundles are arranged on the apical surface of the sensory hair cells in the organ of Corti. The stereociliary bundles are made up of kinocilium, which is situated on the abneural side, and an actin-based stereociliary bundle arranged in a staircase fashion, heading in the direction of the abneural side. The function of the inner ear depends upon the orientation of hair cells because it enables the hair cells to detect the direction of mechanical stimuli. The first indication that PCP controls the polarization of sensory hair cells came from circletail and looptail mice [118]. They observed disruption of the exact arrangement of the stereociliary bundle in both mutant mouse models. Analogous discoveries were also noted in Drosophila PCP mutants, exhibiting haphazard organization of ommatidia. Experiments using animals lacking the PCP core proteins (Celsr1/Flamingo, Ptk7, Fz3/Fz6, and Dvl1/Dvl2) revealed that the organ of Corti had abnormal hair bundle orientation [85,119,120,121]. Wnt is involved in the orientation of stereociliary bundles, as demonstrated by an in vitro investigation that found Wnt ligands to be permissive factors for this orientation [122]. In Drosophila, Fz and Stbm are situated at the opposing poles of sensory cells in the inner ear, whereas Vangl2 and Fz receptors (mFz6 and mFz3) are found on the same side [88]. Fz6 and Fz3 exhibited impressive redundancy, whereby one Fz can make up for the other’s absence at the protein level [88]. A study was performed on a conditional knockout mice model and showed the connection between IFT mutation (transport protein-88) and PCP in the polarization of sensory hairs [123]. One study concluded that stereociliary bundles are unable to react to the positional signals given by PCP signaling in the absence of the cilium [124]. Subneumery, ectopic, and supernumerary hairs are indicative of various deformities associated with PCP signaling when Notch–Delta transmission is disrupted [125,126]. The appearance of an abnormal cochlea (wide and short) was linked to a malfunctioning convergent extension mechanism [127].

## 12. Skin

Studies revealed the relationship between skin biology to epidermal development, hair orientation, epidermal wound healing, and stem cell biology of skin and PCP [61,128,129,130,131]. The preservation of the polarization state throughout epidermal development depends on the distribution of genes of core PCP in a particular cellular compartment [132]. Defective cuticle formation in Drosophila was found to be associated with genetic alteration in genes of the core PCP, resulting in altered formation of the skin barrier [66]. It has been demonstrated that PCP signaling disruption controls both fur and sensory hair cell polarity in mammals [129]. Wild-type mice have a well-groomed hair coat in which each hair is oriented caudally; however, PCP mutant mice (Celsr1 or Fz6) exhibited whorl-like hair patterns concentrating on specific regions of the head, limbs, and body [66,133]. Data from investigations showed that disruption of asymmetric cell division leads to alteration in terminal differentiation, barrier formation, and epidermal stratification because it regulates Notch-mediated differentiation of the epidermis [134,135]. Grhl3 (a novel vertebrate inducer of PCP) is involved in various functions associated with skin biology, including epidermal wound healing and the formation of the skin barrier [136,137,138]. Grhl3 participates in genetic interactions during the epidermal wound healing stage with genes of core PCP (Celsr1, PTK7, Vangl2, and Scrib1) [137]. Grh, a Grhl3 homolog, is involved in various functions, including cuticle repair and epidermal development in Drosophila [138,139,140].

## 13. Other Signaling Pathways

Crosstalk between various signaling pathways during embryonic development is crucial for minimizing multiple signaling pathways to attain functional and anatomical complexity. PCP signaling functions by regulating other signaling channels through interaction; for instance, Wnt mediates the non-canonical Wnt signaling pathways and acts as a ligand for subsets of Fz proteins [25]. Several lines of reports demonstrated the relationship between Fat signaling and PCP signaling at the expression gradient level of Fj and Ds [30,48,51]. Dvl, a multi-domain protein found in vertebrates, mediates both the non-canonical and canonical Wnt signaling pathways [20]. There have been reports of interactions between apicobasal determinants and PCP proteins [141].
(i)**Wnt Signaling:** The Wnt protein activates the PCP signaling pathway through the activation of a transmembrane protein called Fz [9]. Data from multiple vertebrate investigations showed that Wnt11 and Wnt5a are involved in the induction of PCP [142,143,144,145]. Wing experiments on *Drosophila* reported that dWnt4 and Wg exhibit a crucial instructive role in positioning the PCP axis [146]. It has been demonstrated that Wnt5a interacts with complex receptors in PCP signaling that contain Ryk, Fz, Ror2, and Vangl2 [146,147,148]. In vertebrates, Wnt5a and Wnt11 also play an instructive role in activating PCP [147,149]. Studies on mutants (silberblick and pipetail) observed that mutations in Wnt11 and Wnt5a exhibit defective A-P axis (shortened and broadened) because of disrupted convergent extension movements, indicating that a Wnt signaling pathway is needed to control the convergent extension process via PCP [90,147]. Wnt signaling is not only involved in the regulation of PCP-mediated convergent extension processes but also mediates limb elongation by regulating PCP. Outcomes obtained from the Wnt5a mutant mice model showed that Wnt5a is implicated in the regulation of PCP-mediated limb elongation [150]. Genetic studies on the Wnt5a null mice models found that Wnt5a is very crucial for the establishment of PCP in the developing limb [151].(ii)**Hippo Signaling:** Hippo signaling is considered as a key regulator of organ size by regulating cell apoptosis and proliferation in mammals and flies [152]. Several lines of experimental evidence have reported the crosstalk between Hippo and PCP signaling pathways [152,153]. The relationship between Hippo and PCP signaling may be crucial for regulating the orientation of cell division during embryonic development, which is crucial for defining the form of tissues [154]. Ft, a proto-cadherin molecule of the Hippo signaling pathway, is needed for proper PCP in multiple developing tissues in Drosophila like fate choice positioning during the development of ommatidia, hair positioning in the abdomen and wings, and larval denticle orientation [30,48,155,156]. Through the regulation of Ft activity, patterned Ds serve as a cue for PCP orientation and the formation of imaginal discs in Drosophila [30,49,157,158]. While depletion of Fj and Ds causes partial changes in the growth of wings, PCP is engaged in the normal development of wings with uniform expression of Fj and Ds [50,51,153]. The study on the Ft mutant showed that the absence of the ECD (extracellular domain) greatly improved the PCP defects in the abdomen and wings of the ft mutant [157]. An in vivo study conducted on mammals observed that depletion of Fat4 is associated with PCP defects owing to loss of Ds1 [158,159,160]. It has been proposed that there is an overlap between PCP and Hippo functions since the Hippo pathway controls Fj expression [152]. Atrophine/Grunge (a transcriptional co-regulator) is also involved in the regulation of PCP by interacting with the ICD (intracellular domain) of Ft [52].(iii)**Notch Signaling:** Notch signaling is a highly conserved signaling cascade involved in the coordination of multiple developmental processes [161,162,163]. It has been demonstrated that Notch signaling is regulated by PCP, like in the development of *Drosophila* legs and eyes [163,164]. Studies observed the interplay of Notch signaling and PCP in ommatidial rotation in the eyes of insects [164,165,166]. One study on PCP mutant legs showed that ectopic Notch activity is associated with ectopic joints, indicating that PCP regulates Notch signaling [167]. In the *Drosophila* eye, PCP/Fz signaling determines the R3 fate from the precursor while inducing Notch-mediated signaling in adjacent cells to determine the R4 fate [164,165,166]. Genetic alterations in PCP result in random location of the ommatidial, R3/R4 specification, and related chirality [168].(iv)**Sonic Hedgehog (Shh) signaling:** The complex signal transduction mechanisms that control the finely tuned developmental processes of multicellular animals include the Sonic Hedgehog (Shh) signaling cascade. It also has a significant part in the processes of post-embryonic tissue regeneration and repair in addition to setting the patterns of cellular differentiation that control the creation of complex organs. The development of diverse neuronal populations in the central nervous system is specifically linked to Shh signaling [169]. The Shh signaling pathway involves a series of molecular events that occur when the Shh protein binds to its receptor, Patched (Ptch), relieving its inhibition on another receptor called Smoothened (Smo). This activation of Smo triggers a cascade of intracellular events, ultimately leading to the activation of transcription factors such as Gli proteins. These Gli proteins then regulate the expression of target genes involved in cell fate determination, proliferation, and differentiation. The Shh signaling pathway is essential for the development of various tissues and organs, including the central nervous system, limbs, and organs such as the lungs and gastrointestinal tract. Dysregulation of this pathway can lead to developmental defects and diseases, including various types of cancer. Therefore, understanding the mechanisms of Shh signaling holds great promise for both developmental biology and clinical applications.

## 14. Negative Regulation

It has also been suggested that cytoplasmic proteins may play a role in amplifying asymmetry by repelling Fz- and Vangl-comprising complexes. According to an in vitro investigation, Pk and Dgo have mutually exclusive interactions with the same Dsh domain [40]. Furthermore, it has been observed that overexpression of Pk inhibits Dsh translocation to the cell membrane, indicating that interaction of Pk with Dsh may cause it to shift from the proximal side of the cell [43,170]. The interaction of Dgo with Dsh hampers the association with Pk on the farthest side, resulting in Pk/Dsh asymmetry, followed by the establishment of a positive feedback loop. By limiting the quantity of one PCP protein that may oppose another, modification of PCP protein levels by ubiquitin-mediated degradation also produces feedback [171]. A study conducted on Drosophila showed that the Cullin-3-BTB-E3 ubiquitin ligase complex restricts the Dsh level at cell junctions by regulating Dsh [172]. They also found that a decrease in the Cullin-3-BTB-E3 ubiquitin ligase complex leads to an increase in the level of core PCP proteins and a decrease in asymmetry. SkpA, a component of SCF-E3 ligase, modifies Pk levels by degrading them, which depends upon Vang interaction [173]. One study carried out on a mouse model observed that Smurf-E3 ligases induce the binding of Pk with phosphorylated Dvl by ubiquitinating it, followed by its proteasomal degradation [174]. They also found that genetic mutation in Smurf leads to defective alignment of stereocilia and a convergent extension process. Pk2 expression levels are restricted by the Vangl2 interaction through polyubiquitination in a way that is reliant on Cullin-1 [171].

## 15. Genetic Disorders

An organism’s ability to grow healthily depends on its PCP system, and mutations in proteins related to PCP systems can result in a number of diseases. There is evidence linking some hereditary illnesses to abnormalities in the PCP system [175,176,177,178,179]. Polycystic kidney disease is linked to irregular PCP regulation [175]. According to a number of studies, Vangl genetic mutation causes NTDs to occur [41,65,84]. There has been evidence from experiments that DVl mutation and NTD are related [85,180]. There is a strong correlation between NTDs and numerous other mutations in the key PCP proteins (FZ6, FZ3, FZ1, Scy, Crsh, Stbm, Diver-sin, and Pk1) [63,86,87,88,89,92,93,94,180]. Autism spectrum disorders, or ASD, have been linked to abnormalities in the PCP gene, according to research carried out on mouse models [175]. Disrupted PCP in epithelial cells has been linked to cardiovascular disorders, according to several studies [181,182]. Age-linked macular degeneration and PCP loss in the epithelium were observed to be correlated in one study [183]. Vangl1/2 and Dvl2/3 mutations have been shown to produce cardiac anomalies such as tiny ventricles, double-outlet ventricles, heart looping, retro-esophageal subclavian arterioles, and ventricular septal defects in a number of experimental experiments conducted on looptail mice models [184,185,186,187]. Female reproductive tract abnormalities are partly caused by genetic mutations in the PCP component Scribb gene [188]. A growing body of research has shown the connection between different types of carcinomas and abnormalities in the PCP system, particularly the Fz6 mutation [189,190,191,192,193]. Another investigation found a correlation between genetic changes in PCP genes and intellectual impairment [175].

## 16. CRISPR/Cas9

A modern genetic manipulation engineering technique, CRISPR/Cas9, can be used to edit a gene to create knockout mice models (point mutations, knock-outs, and knock-ins), which can be useful for studying the molecular mechanisms of genetic disorders [194]. There is widespread use of CRISPR/Cas9 technology in a variety of scientific fields, including animal and plant sciences, therapeutics, and medical sciences [195,196,197,198]. Several experimental investigations employed the CRISPR/Cas9 technique to manipulate the gene in various models to explore the role of PCP proteins in embryonic development [199,200,201,202,203]. The CRISPR/Cas9-mediated generation of a Celr1/2 mouse model explores its role in the establishment of plane polarity in the skin [199]. Characterization of the role of proteins in the core PCP system in axonal guidance is explored through the use of CRISPR/Cas9-mediated manipulation of Ror1/2 genes [201]. CRISPR/Cas9 and TALEN (Transcription activator-like effector nucleases)-mediated generation of c21orf59/kurly Knock-in zebrafish models demonstrated the role of c21orf59 in the regulation of polarization and motility of cilia [203]. A recent study studied the role of core PCP genes (Vangl1/2) in NTDs through the generation of knock-ins and knock-outs in *Drosophila* and mice models [202]. One study used CRISPR/Cas9 technology to generate somatic mutation and found a correlation between the core PCP system and polarization in epithelial cells [200]. A wide-spectrum study that investigated the relationship between abnormalities in core PCP proteins and other diseases [204,205,206] made use of CRISPR/Cas9 technology.

## 17. Challenges in Planar Cell Polarity (PCP)

Studies in biochemistry, genetics, and cell biology—three hierarchy-tiered modules—provide data on PCP. A three-tiered hierarchy was first proposed as a result of several discoveries. Since a number of the components are tissue-specific, the core module for controlling them and epistasis—which implies this architectural style—was suggested [44,200,201,202]. Because of the misalignment of core PCP proteins in global mutant wings resulting from eye studies on epistasis, the global module was first defined upstream of the core module [30,49]. This three-tiered hierarchy module between tissue-specific, core, and global modules revealed a linear association [6]. The global module transforms relatively shallow transcriptional gradients into tenuous sub-cellular gradients [6]. While the tissue-specific module recognizes the polarity signals and transforms them into cell fate or morphological asymmetry, the core module simultaneously amplifies the locally aligned polarization and sub-cellular asymmetry [6]. The aforementioned observations point to a linear link between these three modules; yet, there is a novel molecular mechanism that accounts for the information flow between these modules, and the molecular interactions between them are still unknown. The linear three-tiered hierarchy model is challenged by genetic research on the larval epidermis and the polarity of denticles in the adult Drosophila abdomen [157,207,208,209,210,211]. It has been proposed that the linear model is at odds with two significant observations: (i) Compared to single mutants of each module, double mutants generated by combining elements of the core amplification module and the upstream global module exhibit more pronounced polarity abnormalities in the adult adipocytes and larval denticles [157,210,212]. This increase in the mutant phenotype suggests that the core and upstream modules can influence the downstream players concurrently. (ii) It has been shown that overexpressing certain upstream module components in the abdomen can alter the polarity of denticles even in the absence of the core signal amplification module [213], demonstrating a direct relationship between tissue-specific polarization signals and global directional signals. We believe that the linear model is still the most realistic way to depict the connections between the three modules based on the data that has been available up to this point. The disappearance of the linear three-tiered module seems unlikely as long as we have not learnt more about the molecular level of signal transmission between modules.

## 18. Conclusion and Future Perspectives

The PCP signaling pathway, its constituent parts, its relationship to other signaling pathways, and related hereditary problems have all been emphasized in this review. The early death of models due to genetic abnormalities and functional redundancy among genes, however, is likely to result in a great deal of confusion regarding the significance of PCP signaling. Additionally, the cellular information flow between global and core modules is still largely driven by primitive chemical mechanisms. Thus, investigations of the PCP system’s molecular connections, behavior in a variety of cells, and tissue-level effects should be the forefront of future study. In order to identify the precise molecular mechanisms that produce the information cues flowing between global, core, and tissue-specific modules and that will explain how they contribute to the localization of the PCP protein and the final asymmetric result, more advanced integrative mathematical models must be prepared to integrate with biological experimentation.

## Figures and Tables

**Figure 1 jdb-12-00012-f001:**
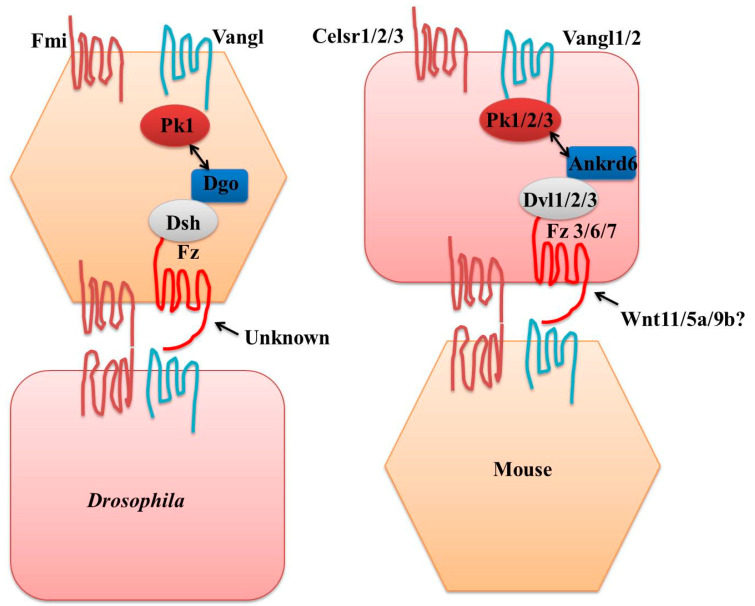
PCP-mediated signaling pathway in *Drosophila* and mouse. Planar cell polarity (PCP) signaling in flies and mice is organized similarly. Orthologous genes (have similar colors) are shown. Observe the heterophilic interactions between the proteins fz/Fzd and vang/Vangl, and the homophilic interactions between the fmi/Celsr pair. While the mechanism of Drosophila frizzled activation is yet unknown, at least some of the ligands in vertebrates are known. Not every paralog of vertebrates has been linked to PCP signaling. Ankrd6, also known as Diversin, is Ankyrin repeat 6 and may be orthologous to Diego.

**Figure 2 jdb-12-00012-f002:**
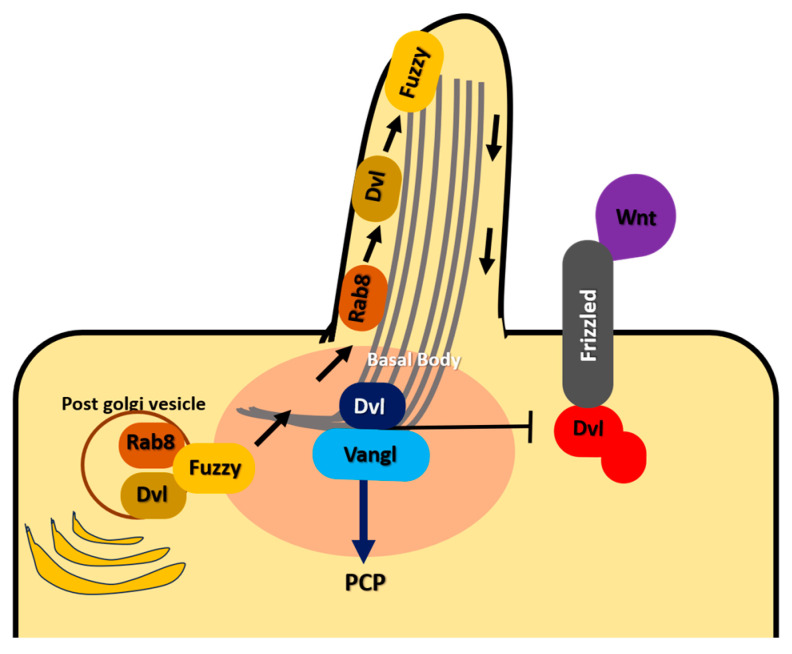
Cilial PCP signaling pathway. Fuzzy arbitrates the relative activity of canonical and noncanonical (PCP) Wnt pathways and transports Dvl to the cilium via Rab8(+) vesicular traffic. Rab8 and Dvl2 must be recruited by Fuzzy to the polarized trafficking route in ciliated cells for both molecules to reach the primary cilium. Dvl2, which is intracellularly accessible for canonical Wnt signaling, is reduced by sequestering it into vesicles. Dvl2 orchestrates basal body polarization and promotes PCP signaling at the cilium of wild-type cells. Without Fuzzy, ciliogenesis is hampered, Dvl2 recruitment to the basal bodies is impaired, and the Rab8-trafficking route is interfered with.

**Figure 3 jdb-12-00012-f003:**
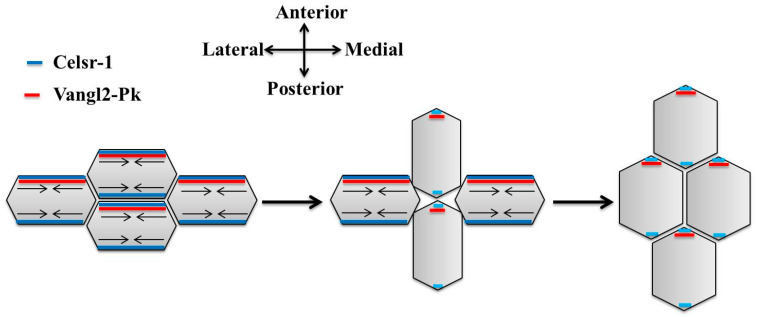
Asymmetric localization of PCP proteins promotes convergent extension motion in neuro-epithelium sheet.

**Table 1 jdb-12-00012-t001:** Components of core PCP system [41].

Drosophila	Vertebrates	Type of Protein
Fz (Frizzled)	Fz3, Fz2, Fz7, Fz6	Extracellular-rich cysteine domain,
		VII-pass trans-membrane receptor
Stan/Fmi (Starry night/Flamingo)	Celsr3, Celsr2, and Celsr1	VII-pass trans-membrane receptor,
		Extracellular cadherin-repeat
Pk (Prickle)	Pk2 and Pk1	PET-domain, Triple-LIM domains,
		Cytoplasmic
Dsh (Dishevelled)	Dvl3, Dvl1, and Dvl2	PDZ, DIX, DEP, Cytoplasmic domains
Vang Gogh/Strabismus	Vangl2 and Vangl1	PDZ-binding domains, IV-pass
		Trans-membrane receptor
Dgo (Diego)	Inv (Inversin)	Ankyrin-repeats, Cytoplasmic

## Data Availability

Not applicable.

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
