# Peer review of "Planar Cell Polarity Signaling: Coordinated Crosstalk for Cell Orientation"

_jdb, 2024, doi:10.3390/jdb12020012_

Round 1
Reviewer 1 Report
Comments and Suggestions for Authors
Manuscript ID: jdb-2869778
Type of manuscript: Review
Title: Planar Cell Polarity (PCP) Signaling: The Coordinated Crosstalk for
Cell Orientation
Journal: Journal of Developmental Biology
This review covers the PCP signaling components, related physiological processes, connection with other signaling pathways, and associated diseases. It is comprehensive overall with relevant references, but several major limitations should be addressed for the publication.
Major concerns.
1. There exists inconsistency regarding the PCP effectors covered in this review. In the introduction section, an overall review of PCP effectors (Fuz, Intu and Wdpcp) is missing, as the authors mentioned these genes/proteins in the later sections, e.g. neural tube closure, wing hair, and neural tube defects.
2. As the authors reviewed the connection between cilia and PCP signaling regarding polarity and orientation, Shh signaling should be covered in the “other signaling pathways” section as it is critically relevant to cilia and PCP effectors (CPLANE complex).
3. In the “other signaling pathways” section, Fz-Wnt coupling is canonical Wnt or PCP signaling-specific. Rather, Dvl or Lrp6 might be more relevant to cover in conjunction with canonical Wnt and PCP signaling. The author should clarify this.
3. Some references should be replaced with appropriate and relevant research papers.
For example,
P132: Reference (9) should be replaced with the original research article showing the role of Fmi.
p.196-198: Reference (9) is improper for referencing the prevalence of neural tube defects.
4. Some of the statements need to be clarified and refined.
For example,
p. 201: Encephalocele is not the neural tube closure defect, as it is considered to occur post-neurulation stage. In addition, its affected area is not limited to the forebrain or midbrain. The authors should clarify this statement.
P211: Any single genes that solely lead to human NTDs have been identified. Vangl1 is associated with the genetic etiology of NTDs. Authors should specify the statement.
P446: Fuzzy and Inturned are not core PCP proteins.
Reviewer 2 Report
Comments and Suggestions for Authors
In the presented manuscript, Kacker and colleagues review PCP signaling pathway involved in the regulation of cellular and tissue polarity during various biological processes. The topic of the work is extremely relevant in connection with the unresolved problems of developmental patterning, morphogenesis, and differentiation in normal and pathological conditions. The strength of the work must be recognized as the breadth of coverage of the molecular biological machinery of PCP. The authors focus their attention on functional aspects of the involved proteins, their interactions, and phenotypic outcomes of mutations.
General comments
Unfortunately, the authors do not put forward conceptual ideas and do not conduct a deep analysis of the problem. All that is stated in the manuscript is a listing of a large number of facts. In general, the text of each section is difficult to understand and completely lacks cohesion. A very big drawback is the lack of hierarchy in sections. There are no paragraphs within the sections, almost no development of thought and narrative logic. There are often repetitions of stories about the same processes. For example, about convergent extension, the essence of which is not disclosed. It seems that the authors themselves are not very well versed in the field of embryonic development, and therefore cannot provide a comprehensive discussion of the topic. The models discussed are limited almost exclusively to Drosophila and mouse, which does not correspond to the current knowledge on PCP signaling. The illustrative material is insufficient and relates poorly to the texts and legends. The table shows errors in the spelling of genes (Prikled instead of Prikle, Strbismus instead of Strabismus), there are unfinished phrases in the text (lines 143, 258) and other ridiculous errors. Overall, the manuscript makes a poor impression and, in my opinion, cannot be corrected to a publishable state.
Reviewer 3 Report
Comments and Suggestions for Authors
To get an understanding of how cell shape and - as a consequence - tissue structure are grounded on molecular-genetic interactions and networks is a topic of utmost importance. In hoping to get an expert upgrade of my knowledge on this fascinating topic, I have agreed to prepare this reviewing job. I am afraid that after studying carefully this MS, my knowledge on PCP has only marginally improved. There is a very long list of nearly 200 citations, yet their integration to make a condensed, clearly intelligible story is not satisfactorily achieved. The wealth and diversity of cited data on PCP functionings often is only listed without much commenting. How shape/shaping of cells is brought about by a simple - which of course means a simplified - scheme of molecular PCP mechanism(s) is not resolved. Besides improving the text, a few more figures could also be helpful to give the text more flesh and comprehension; e.g., on sections of Positioning, Cilia and Centrosome, Cochlea, etc.. In other words, this review does not yet deliver on what the Abstract had promised me.
Major comments:
- l. 32: it remains unclear where Introduction ends: does section on „History“ still belong to Intro? The division of the entire MS into headings and sub-headings should be clearer;
- l. 41/42: two chief signaling pathways for PTP are mentioned: throughout the following entire text, I miss a clear presentation of, and distinction between these two pathways; for instance, where is it in Fig. 1 (or: which one of the two is presented in Fig. 1)?
- l. 52-57 are nearly exactly a repetition of l. 23-28: improve wording;
- l. 58ff: Para on History is not well organized (if not to say chaotic); for instance, one could begin with „The pioneering work of Lawrence…“: reorganize.
- l. 71ff, Para on Molecular/Cellular…: this entire section again is not well organized, to be frankly, this is a mess. It is very confusing for a reader who is new to this field. It is an assembly of individual, non-coherent notions. It needs much more intellectual structuring. Begin with the very basic notions and then go further into details. For instance, raise the question/topic which you are going to discuss (even if it was said in Intro - say it briefly again); begin with figure 1 and go step by step through the possible pathways/actions/remarks. Details on Fz actions should come only as you go along the pathways, introduce the topic of this section (what was said is not sufficient).
- l. 71ff: the two models on Fz (l. 73ff) could come before the details on Fz.
- l. 114/115: Fig. 1 needs improvements, for instance: 1. Why are the two types of cells (yellow and red) shown in reversed (upper-lower) order for the two model animals, is there a reason for this? 2. What is meant by black arrows in both model animals? …is the particular Wnt-type unknown, or, is it unknown whether a Wnt is there at all in either model? Why not just put into the schematic that there is a Wnt protein, and let it open which type could it be (such details could come in table 1)?
- l. 116: legend to Fig. 1 is too short, e.g., explain the general scheme of PCP; it needs comments; abbreviations should be explained;
- l. 117-120: where are CELSR and FZD, VANGL, FZD , etc. in Fig. 1? If you change designations of proteins or genes, this needs to be explained in legend to figure or in text (table 1 does not fully resolve this problem, since, for instance, for Drosophila Vang in table is denoted as Vangl in Fig. 1: which one is the correct name?): all this contributes to my/the reader´s confusion.
- After l. 122 follows Table 1. After the table itself, where is legend to table 1, does it begin in l. 125, or is l. 125 the continuation of the flowing text: I guess so, but this is all very confusing and frustrating!
- l. 162: this paragraph on Tissue morphogenesis is well written, much different in style as compared with the text before, probably due to another co-author? I advise this co-author to help to improve the text above accordingly.
- l. 194ff: Para on NTDs is good;
- l. 233ff, dto. on Tissue regeneration;
- l. 439ff: this paragraph consists of a long list of individual studies of genetic disorders, which are factually not connected with each other: reading becomes annoying; improve style of writing.
- l. 479ff: this interesting section is difficult to conceive for an outsider: a schematic figure showing the different models could help.
Minor comments, language and typos:
- l. 16, type: The planar cell polarity (PCP) system…;
- l. 71-139: this section is not well written, at times quite chaotic, see below;
- l. 37 type …and epithelia…;
- l. 51: type …and homeostasis of tissue, and abnormalities…;
- l. 55: sentence following line 55ff is nearly the same as in l. 26ff: plse. change;
- l. 60: …and the eye…is a repetition of what has been said before in the same line;
- l. 99: type …Dominating non-…;
- l. 105;: type …symmetry, and planar…;
- l. 122: where is end of legend to Fig 1? Complete legend!
- l. 117ff: distinction between Drosophila and Vertebrate components remains unclear; for instance, type …core proteins of Fz/PCP signaling processes in Drosophila (Fz-Vangl…in vertebrates)…;
- l. 143, text unclear: …(CAM) The discovery of Fmi….: correct;
- l. 195: „NTD“ does not stay for „neural tube closure“ as said in brackets: I guess the „D“ stands for defect? Correct!
- l. 198: sentence „Approximately…“ is typed in different format: correct! (similarly l. 293ff, l. 459);
- l. 256: …mediolateral… - to what? Substantive is missing;
- l. 298: style … protrusion protrudes…: change;
- l. 342: should it read …stem cell biology…?
- l. 363, too long sentence: type … functions. For example, Wnt…;
- l. 408: type …Notch signaling is…;
- l. 440: type …of organisms, and genetic…;
Round 2
Reviewer 1 Report
Comments and Suggestions for Authors
Many points were addressed in the revised manuscript, but there is still misleading information, which should be addressed.
Lines 150-: The description of Shh signaling should be moved to the "other signaling pathway" section. Also, more comprehensive information regarding the connection to PCP signaling should be included.
Line 257-258: The information on NTD prevalence is incorrect. The reference for NTD prevalence should be replaced. e.g. https://pubmed.ncbi.nlm.nih.gov/36882610/
Author Response
Dear Reviewer,
Thank you for your time to review our manuscript. Your suggestions are incorporated (File is attached for your kind perusal)
Thank you once again.

Reviewer 3 Report
Comments and Suggestions for Authors
This MS has been improved.
Author Response
Dear Reviewer,
Thank you very much for taking the time to review this manuscript. Your positive comments about the manuscript is highly appreciated.